# Waste Not, Want Not;
# Recycled Gumbel Noise Improves Consistency in Natural Language Generation

**Damien de Mijolla,  Hannan Saddiq,  Kim Moore**
Faculty Science Ltd
`damien.de-mijolla@faculty.ai`

## Abstract

Consistency in the output of language models is critical for their reliability and practical utility. Due to their training objective, language models learn to model the full space of possible continuations, leading to outputs that can vary significantly in style, content, and tone, even for similar inputs. To address this, we propose a novel decoding algorithm that enhances response consistency across different prompts with no degradation in response quality. By incorporating a latent variable into the next-token sampling process based on the Gumbel reparametrisation trick, our method outperforms standard sampling by up to 10% across semantic and stylistic consistency benchmarks. Additionally, our approach integrates seamlessly with existing sampling methods with negligible computational overhead, providing a practical solution for improving the reliability of language model outputs.

## 1   Introduction

In recent years, state-of-the-art language models (LMs) have demonstrated remarkable performance across a wide range of benchmarks, often rivaling human capabilities in tasks such as translation, summarization, and question-answering [1; 2]. However, these advancements have not always translated into practical usefulness for real-world applications, where reliability and consistency are crucial [3].

One of the primary challenges is the inconsistency of these models' responses, which can vary significantly in style, factual accuracy, and tone [4]. This inconsistency, a byproduct of the probabilistic nature of language model training, can lead to a range of issues, including reduced trust in outputs, exposure to more diverse failure modes and less reliable behaviour [5].

Although traditional methods (e.g the use of random seeds) can be applied to introduce determinism in natural language generation, ensuring identical responses for identical inputs, they do not help ensure similar responses when inputs are similar. In practice, due to the richness of language, input queries can often be reworded in many ways while retaining their meaning. To achieve greater consistency, it is desirable for the model to generate similar responses across all these variations.

In this paper, we investigate whether next-token sampling procedures can be modified to enhance consistency across different prompts. Our main contributions include:

1. We propose a simple, computationally inexpensive sampling procedure that (i) can be applied to any model, (ii) does not require any additional training, and (iii) has negligible impact on inference costs. We also ensure that the probability of any individual response is unchanged and so does not compromise response quality.

38th Conference on Neural Information Processing Systems, Safe Generative AI Workshop, (NeurIPS 2024).

2. We also leverage an auxiliary approach to further improve consistency between model responses using distributional ensembling, which can be applied in conjunction with our aforementioned sampling procedure.

3. We investigate the performance of our approach against standard sampling across a number of benchmarks covering semantic and stylistic similarity, across a number of different models.

In particular, we highlight that our combined sampler outperforms standard sampling across all benchmark suites and models tested, by up to 10% in some cases.

## 2   Related works

**Decoding approaches**   Language model decoding strategies can be broadly classified into two categories: optimization and sampling-based approaches [6]. Optimization-based approaches, such as greedy decoding and beam search [7; 8], frame text generation as an optimization problem, searching for sequences that maximize a specific metric such as probability, whereas sampling-based approach incorporate stochasticity into the next-token selection process. Optimization-based approaches are typically perceived as yielding less engaging but more accurate responses and so are often favoured for closed-ended tasks expecting a fixed answer [9]. However, recent work has put into question the greater accuracy of their responses [10].

In contrast, sampling-based approaches are usually preferred for open-ended tasks, as they typically yield more engaging answers [11; 6]. Our proposed method falls within this category. Many existing methods in the literature, such as nucleus sampling and mirostat [9; 11; 12], aim to improve text generation quality by directly modifying the probability distribution from which tokens are sampled. We consider these methods, which directly alter the next-token distribution, as complementary to our approach, which maintains the next-token distribution and instead modifies the joint distribution over responses.

Our approach is methodologically most closely related to methods [13; 14] which also adjust the joint distribution of sampled responses. However, while these methods aim to maximize response diversity—an advantage when ensembling multiple responses as done in self-consistency voting [15]—our approach is distinct in its focus on minimizing response diversity to achieve more consistent outputs.

**Self-Consistency**   Language models lack robustness to prompt variations [16; 17] and give contradictory responses in such cases, motivating the need for enhanced self-consistency. Self-consistency in language models has been studied from many different angles, but usually with a focus on factual rather than stylistic consistency. Prior work has proposed a number of fine-tuning approaches for increasing self-consistency, including fine-tuning approaches for increasing the ability of language models to respond consistently to paraphrases of questions [17; 18], and approaches for correcting model contradictions using a factor graph over beliefs [19].

Our approach is methodologically orthogonal to previous approaches for enhancing self-consistency. Previous work has relied on fine-tuning which not only is more cumbersome to implement but also modifies the raw next-token probabilities, potentially affecting responses in unforeseen ways or contributing to catastrophic forgetting.

Since our approach only modifies the joint distribution over responses without modifying the next-token probability distribution, it does not suffer from the same issues, and comes with principled guarantees around maintaining the model's original response style and quality. Additionally, it enhances *all* aspects of self-consistency, not just factual consistency of responses.

## 3   Problem statement

Let $X$ be a language model prompt composed of a sequence of tokens drawn from a vocabulary of size $N_v$, and let $\pi_\theta$ be a language model trained on the task of next-token-prediction. For the remainder of the paper we denote a forward pass through the language model by $h_t = \pi_\theta(X, Y_{1:t-1})$ where $h_t \in \Delta^{N_v - 1}$ is a probability distribution over the token vocabulary and $Y_{1:T}$ is the full response obtained by auto-regressively applying the language model with the next token at each step sampled

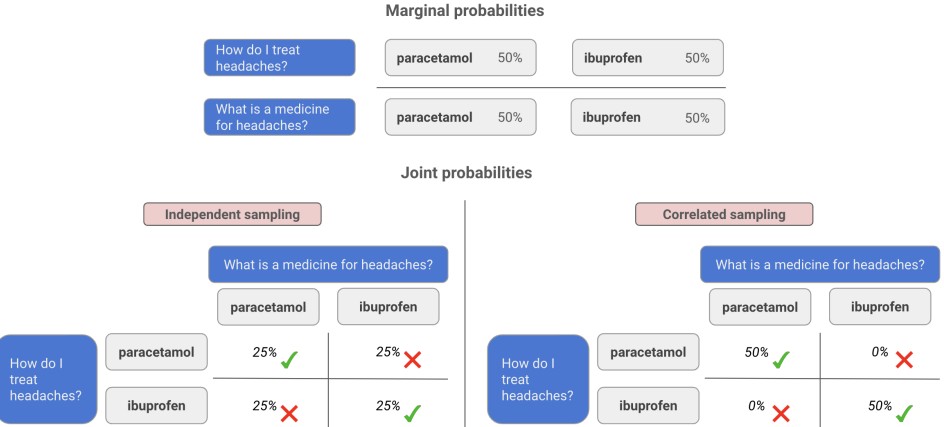

Figure 1: Motivating toy example highlighting the aim of our approach. Even when language models yield similar probability distributions over responses, responses sampled independently can be inconsistent or contradictory due to the inherent stochasticity of sampling. By generating responses in a correlated manner it is possible to alleviate inconsistencies across responses while still respecting the marginal probabilities of each response. In this paper we propose, Gumbel Consistent Sampling, an approach for increasing response consistency through drawing correlated responses, by conditioning all responses on a shared latent variable, that is robust to differences between probability distributions over responses.

from the categorical distribution parameterized by the model, $Y_t \sim \text{Cat}(h_t)$. In what follows, we use a subscript to represent position in a sequence, and a superscript to represent the token index. So for example, $h_t^i$ represents the probability of sampling token $i$ at position $t$.

Suppose that $U$ is a different prompt that is semantically similar to $X$ for which we generate a response $V = V_{1:M}$. Motivated by the inconsistency of LM responses, our goal is to modify the LM sampling procedure in a way that increases the similarity between responses $Y$ and $V$ according to some yet-to-be-specified notion of similarity. Furthermore, we focus on sampling approaches that modify the joint probability of responses $p(Y, V)$ without affecting the marginal probability of individual responses, $p(Y)$ and $p(V)$, to guarantee that quality of the original responses is maintained.

## 4 Approach

Our proposed sampling approach, motivated in Figure 1, modifies the joint probability distribution over responses by introducing a latent variable $g$ to the sampling process. Conditioning the generation of distinct responses on a common realisation of this latent variable introduces a statistical dependency between them. Generating responses with greater similarity can then be straightforwardly done by conditioning the generation of all responses on a common realisation of the latent variable, that is to say to sample $Y \sim p(Y|X, g)$ and $V \sim p(V|U, g)$.

To ensure efficacy of the approach, we design the latent variable in such a way that conditioning responses on a common value of the latent variable makes responses as similar as possible. To ensure the preservation of the probability distribution parameterized by the language model, we sample the latent variable from a probability distribution $g \sim p(g)$ such that marginalising over the latent variable recovers the original distribution over responses, $\mathbb{E}_g[p(Y|g)] = p(Y)$.

To construct a latent variable with the above properties, we employ the reparametrization trick for categorical distributions. Introduced for normal distributions in [20] and extended to categorical distributions in [21; 22; 23], the reparametrization trick is a procedure that refactors the sampling from a distribution into a deterministic function of the parameters and a draw from some independent noise with a fixed distribution. For a categorical distribution with parameters $p^1, ..., p^{N_v}$, this can be cast as first drawing random noise $g = (g^1, ..., g^{N_v})$ where each $g^i \sim G(0, 1)$ is independently drawn from the Gumbel distribution [24] and selecting a category $k$ according to $k = \arg\max_i(\log p^i + g^i)$.

**Theorem 4.1.** *Suppose we have two different categorical distributions parametrized by $p^1, ..., p^{N_v}$ and $q^1, ..., q^{N_v}$. Define a joint distribution over pairs of categories $(Y, V)$ by defining*

$$Y = \arg\max_i(\log p^i + g^i), \ V = \arg\max_i(\log q^i + g^i), \tag{1}$$

*where $g^1, ..., g^{N_v} \sim G(0, 1)$ are independent. We have that*

$$P(Y = k, V = k) = \frac{p^k q^k}{p^k q^k + \sum_{i \neq k} \max\{p^i q^k, q^i p^k\}}.$$

Theorem 4.1 (proved in Appendix A) shows that interpreting the Gumbel noise as a latent variable and conditioning sampling events on the same realisation of this latent variable increases the probability of selecting the same category with both distributions compared to sampling from each categorical distribution independently, with identical sampling outcomes in the limit where $p$ and $q$ become identical.

Since generating a response using a LM consists of successive draws from categorical distributions, the above idea can be applied to language modelling in order to increase the token overlap across distinct responses. Indeed, we can generate ahead of time a sequence of independent Gumbel latent vectors, $g_{1:t}$, one for each position in the sequence up to the maximum sequence length, and sample each token using the Gumbel latent vector assigned to that position in the sequence when generating a response. That is to say, drawing $Y_t \sim p(Y_t|h_{y,t}, g_t)$ and $V_t \sim p(V_t|h_{v,t}, g_t)$, where here we denote by $h_{y,t}$ and $h_{v,t}$, the next-token probabilities obtained by running the language model on the context-up-to-now (i.e. $h_{y,t} = \pi_\theta(X, Y_{1:t-1})$, $h_{v,t} = \pi_\theta(U, V_{1:t-1})$ ). We refer to the above approach as **Gumbel Consistency Sampling, GCS**.

This sequential Gumbel sampling approach increases similarity of responses by increasing the rate at which identical tokens are generated at fixed positions in the sequence $p(Y_i = k, V_i = k)$ but has the limitation of not increasing the co-occurrence across sequence positions $p(Y_j = k, V_i = k)$. We expect that two similar responses are likely to contain some of the same tokens, but likely in different positions, so it would be advantageous for our final sampling approach to reflect this.

Introducing such an inter-position correlation in sampling outcomes across sequences is made challenging by the requirement of conditional independence between sampling steps. Indeed, to respect the LM's probability distribution, it is necessary for sequential sampling steps to be independent of each other, i.e. for $p(Y_{t+1}|X, Y_{1:t}) = p(Y_{t+1}|h_{t+1}) = \text{Cat}(Y_{t+1}; h_{t+1})$ which prevents the direct reuse of Gumbel samples across sequence positions.

The procedure denoted in Algorithm 1, which we henceforth refer to as **Gumbel Consistency Sampling with Recycling, (GCSwR)**, respects this property and thus recovers the correct marginal distribution over responses, which we prove in Appendix B.1.

---
**Algorithm 1** Gumbel Consistency Sampling with Recycling (GCSwR)
---
**Input:** Context $X$, sequence length $T$, language model parameters $\theta$
**Output:** Generated token sequence $Y_{1:T}$
 1: Initialize $g \sim G(0, 1) \in \mathbb{R}^{N_{\text{vocab}} \times T}$ and $c = [0, 0, \ldots, 0] \in \mathbb{R}^{N_{\text{vocab}}}$
 2: **for** $t = 1$ to $T$ **do**
 3:     $h_t \leftarrow \pi_\theta(X, Y_{1:t-1})$
 4:     $k \leftarrow \arg\max_j \left( g_{c_j}^j + \log h_t^j \right)$
 5:     $Y_t \leftarrow k$
 6:     $c_k \leftarrow c_k + 1$
 7:     **for** each $i \neq k$ **do**
 8:         $g_{c_i}^i \leftarrow Q\left( \frac{Q^{-1}(g_{c_i}^i)}{Q^{-1}\left(g_{c_k}^k + \log h_t^k - \log h_t^i\right)} \right)$ $\{Q(\cdot)$: Gumbel quantile function$\}$
 9:     **end for**
10: **end for**
11: **return** $Y_{1:T}$

---

This approach relies on the observation that awareness of the value of a token $Y_t$ only reveals the value of $\arg\max_j(\log h_t^j + g_t^j)$ rather than fully revealing $g_t$. From one sequence position to the

next, the recycling procedure involves resampling a new Gumbel noise value for this position in the Gumbel latent vector, but recycling (a rescaled version of) the existing Gumbel values for every other position in the vector. In practice, we generate all Gumbel samples for the resampling ahead of time so that we may use the same sequence of Gumbel vectors independent of each other for each response.

The standard procedure for autoregressive token sampling, which is equivalent to independent sampling of a new Gumbel latent vector for every sequence position and every sequence, acts as a baseline for subsequent experiments, and is denoted as **Independent Sampling, (IS)**.

## 5  Ensembling semantically similar responses

A complementary approach to enhance consistency between responses given semantically similar prompts is to reduce the impact of semantically irrelevant prompt attributes on the next-token probability distributions, which can be achieved by increasing the similarity between the sampling distributions.

In our experiments, we explore sampling tokens from an ensembled probability distribution over semantically equivalent prompts as a means of minimising impact of semantically irrelevant prompt variations on responses. Specifically, we generate semantically equivalent variations of the user prompt by asking a separate LM (`gpt-4o mini`) to rephrase the prompt. We then run the target LM separately on all of the prompts, producing a set $\{P_i\}$ of next-token probability distributions. We then sample from an ensembled distribution, ensembled using the following formula:

$$Q^j = \frac{1}{Z} \prod_{i=1}^{n} (P_i^j)^{\frac{1}{n}} \tag{2}$$

where Z is the normalisation constant that ensures $Q$ defines a valid probability distribution function:

$$Z = \sum_j \prod_{i=1}^{n} (P_i^j)^{\frac{1}{n}}$$

This formula corresponds to selecting the categorical distribution that minimizes the average forward-KL divergence over all next-token probability distributions (see Appendix D). We found that direct averaging (which can equivalently be shown to minimize the reverse-KL distribution) tended to generate worse-quality responses due to at times sampling tokens that were only high-probability for a subset of question rewordings.

Note that, contrary to our proposed Gumbel sampling approach, ensembling comes at a cost of additional inference-time compute and also modifies the language model probability distributions. We highlight that ensembling can be applied in conjunction with any of the three samplers discussed in section 4, and we investigate the performance of each sampler with and without ensembling in our experiments.

## 6  Experiments

In our experiments, we empirically demonstrate the utility and limitations of GCS and GCSwR. We begin by quantifying the utility of the procedure for enhancing semantic similarity of responses, and highlight a number of stylistic dimensions of text along which Gumbel sampling improves consistency. Details for reproducing experiments are shown in Appendix E.

### 6.1  Semantic similarity

We start by quantifying the improvement in the semantic similarity between responses for semantically equivalent queries by using our Gumbel sampling variants (GCS and GCSwR). To measure semantic similarity, we use E5$_{\text{mistral-7b}}$, a specialised state-of-the-art model trained specifically on the task of semantic similarity [25].

We create semantically equivalent pairs of questions for evaluation by randomly sampling 300 questions from the Alpaca dataset [26] — a popular human-preference dataset - and rephrasing

them using `gpt-4o mini`. We then generate responses to the original and rephrased version of each question using `Meta-Llama-3-8B-Instruct`, `Meta-Llama-3-8B`, `Mistral-7B-v0.1`, `Llama-2-7b-chat-hf`[27; 28; 29]. In all cases we sample from the raw unmodified next-token probabilities predicted by the language models (i.e. temperature of 1) and for Gumbel sampling, we resample the Gumbel latent vector for each pair of questions such that responses are correlated within but not between pairs.

The aggregated results, shown in Table 1, demonstrate that the most performant sampling scheme tested (GCSwR with ensembling) significantly increases response similarity to semantically equivalent questions across all models considered, by more than 10% when compared to the baseline in some cases. We note more pronounced enhancements from Gumbel sampling for unaligned models like Mistral and Llama3 Base, which we hypothesise is caused by their lower base semantic similarity compared to their instruction fine-tuned counterparts.

Table 1: Model results by sampler type. Scores shown as mean±std.err with std.err obtained from 3 independent runs. Bold indicates highest scores for each model in both ensembling categories.

| Model | Sampler | Without Ensembling | With Ensembling |
|---|---|---|---|
| Llama2 Chat | IS | 86.34±0.07 | 87.56±0.29 |
| | GCS | 88.28±0.10 | 90.26±0.27 |
| | GCSwR | **88.61±0.15** | **90.38±0.25** |
| Mistral | IS | 72.00±0.27 | 72.34±0.93 |
| | GCS | 78.55±0.22 | 81.17±0.77 |
| | GCSwR | **80.94±1.05** | **82.74±0.81** |
| Llama3 Instruct | IS | 85.61±0.18 | 86.90±0.16 |
| | GCS | 86.81±0.46 | 89.01±0.35 |
| | GCSwR | **87.37±0.27** | **89.68±0.08** |
| Llama3 Base | IS | 71.23±0.41 | 71.46±0.70 |
| | GCS | 76.68±0.80 | 78.71±0.82 |
| | GCSwR | **80.10±0.80** | **82.04±0.81** |

## 6.2 Semantic similarity as a function of temperature

Next, we investigate how the effectiveness of GCSwR varies with sampling temperature. We compare the semantic similarity metric on the Alpaca dataset as a function of temperature in Figure 2 with IS as a baseline, without using ensembling in both cases. GCSwR improves the semantic consistency of responses across all temperatures, except temperature 0, where the model probabilities with and without GWSwR become identical due to the fully deterministic nature of model outputs at this temperature[1] Example responses for Llama3 models at temperature 0.8 can be found in Appendix G.

It is also interesting to note that although GCSwR improves self-consistency at all non-zero temperatures, the highest self-consistency achieved is with greedy decoding (i.e. temperature 0) which is where both approaches behave identically. However, we caution that this result does not imply that greedy decoding will always be preferable to higher-temperature Gumbel sampling. Using greedy decoding is widely considered to decrease the quality of responses across a number of important dimensions and so model providers typically use non-zero default temperatures [11; 6; 30]. Gumbel sampling offers a way of increasing the consistency of responses without the negative side-effects associated with excessively lowering the sampling temperature. We also note that using Gumbel sampling is much more effective at increasing self-consistency of responses than decreasing temperature, with temperatures needing to be roughly halved in order to match the benefits of using Gumbel consistency sampling.

---

[1]We note that responses can still differ under greedy decoding if several tokens are tied for maximum probability. In experiments this occurred a non-negligible amount of times due to the limited numerical precision of bfloat16.

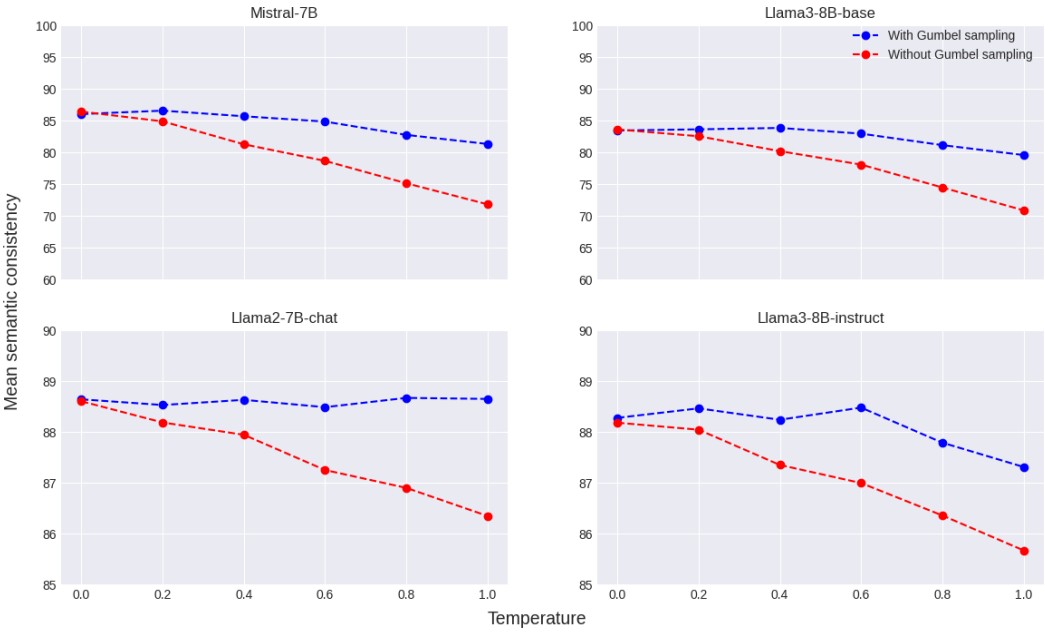

Figure 2: Mean semantic consistency between responses to paraphrased questions as a function of temperature, comparing independent sampling (IS) against GCSwR.

## 6.3 Stylistic similarity

In this section, we study Gumbel consistency sampling's ability to enhance stylistic consistency across several distinct stylistic dimensions, evaluating GCSwR without ensembling using `Mistral-7B-v0.1`.

We conduct our experiments on two datasets: Code-Alpaca and Aleatoric-List. The Code-Alpaca dataset [31] consists of coding-related questions, from which we select a subset of 20 random questions that are agnostic to programming languages. For this dataset, we assess stylistic consistency based on several factors: whether the response contains a code snippet, whether the response starts directly with the code snippet or begins with freeform text, whether the code snippet includes comments, and the programming language used in the response (such as Python, JavaScript, or C++).

The second dataset, Aleatoric-List, is a synthetic dataset we created containing 20 questions that ask for five different items fitting a specific category. An illustrative example question is "Give me the names of five capital cities in Europe." For this dataset, we evaluate stylistic consistency based on whether the answer is terse, whether it contains bullet points, and whether these bullet points are numerical.

To evaluate stylistic consistency along each dimension, we begin by generating 100 Gumbel latent vectors. Then, for each Gumbel vector, we generate a response to all questions in the dataset which we classify along each of the stylistic dimensions through prompting `gpt-4o mini` (with prompts shown in Appendix F). For each factor, we then define the stylistic consistency as the probability that responses to two randomly selected questions share the same label, denoted as $p_{repeat}$. We then compare this probability with the equivalent probability when the responses are generated with our independent sampling baseline (IS).

Let $Z$ be a Bernoulli random variable that denotes whether a randomly sampled response is labelled with a given stylistic dimension, $p(Z = 1) = p$. For IS, $p_{repeat} = p^2 + (1 - p)^2$. However, for GCS and GCSwR, $p_{repeat} = \mathbb{E}_g[p_g^2 + (1 - p_g)^2]$ where $p_g$ denotes the probability of a randomly sampled response generated using Gumbel latent vector $g$ taking value $Z = 1$. These expressions follow directly from the conditional independence of responses generated with a common initial Gumbel latent vector $g$ and generated independently, and additionally from marginalisation over initial Gumbel latent vectors $g$.

Although the estimator $\hat{p} = \frac{1}{n}\sum_{i=1}^{n} Z_i$ is an unbiased estimator of $p$, $\hat{p}^2 + (1 - \hat{p})^2$ yields a biased estimator of $p^2 + (1 - p)^2$. To correct for this bias, we use the following estimator $\frac{N}{N-1}(\hat{p}^2 + (1 - \hat{p})^2) - \frac{1}{N-1}$ which we show in Appendix C to be unbiased.

Table 2: Comparison of GCSwR vs. IS on Stylistic Consistency

| Dataset | Stylistic Dimension | Sampler | |
|---|---|---|---|
| | | IS | GCSwR |
| **Code-Alpaca** | Is Python | 0.67 | **0.73** |
| | Is JavaScript | 0.78 | **0.84** |
| | Is C++ | 0.92 | **0.94** |
| | Contains Code Snippet | 0.71 | **0.81** |
| | Answers Directly | 0.50 | **0.73** |
| | Contains Comments | 0.71 | **0.80** |
| **Aleatoric-List** | Does Not Use Bullets | 0.75 | **0.82** |
| | Uses Numerical Bullets | 0.82 | **0.87** |
| | Terseness | 0.50 | **0.64** |

We show, in Table 2, the results of this experiment, using Mistral-7B to generate responses. Across all stylistic dimensions considered, using GCSwR increases the frequency with which generated responses follow a common style. For many factors, the increase is significant (>10%), showing that Gumbel consistency sampling can have an appreciable impact on style consistency.

## 7  Conclusion

We have introduced Gumbel consistency sampling, a straightforward and computationally inexpensive sampling approach for increasing consistency amongst model responses. The method requires no additional fine-tuning, additional language model calls or apriori knowledge of what prompts will be used, and guarantees responses indistinguishable to those obtained using standard sampling at the level of individual responses. The approach enhances consistency by sampling responses in a correlated manner through the introduction of a latent variable, in a way that increases the token overlap across responses. In our experiments, we find that this approach is not only able to enhance semantic similarity between responses but also stylistic similarity. These results showcase how Gumbel consistency sampling offers a principled quick and easy way of enhancing language model consistency.

Future work could extend the Gumbel consistency sampling to imposing local rather than global correlation to responses. Currently, all responses are globally coupled due to dependence on the same global latent variable, which makes localised adjustments to model behaviour impossible. However, the framework could easily enable for latent variables to be varied locally depending on question specifics, which would enable finer-grain control of model behaviour. Another, promising direction for extending the work could be to treat the Gumbel noise as a learnable task-specific parameter. Such an approach may be especially useful for building stronger model safeguards while preserving general utility.

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

## A  Proof for Theorem 4.1

**Theorem.** *Suppose we have two different categorical distributions parametrised by $p^1, ..., p^{N_v}$ and $q^1, ..., q^{N_v}$. Define a joint distribution over pairs of categories $(Y, V)$ by defining*

$$Y = \arg\max_i(\log p^i + g^i), \ V = \arg\max_i(\log q^i + g^i), \tag{3}$$

*where $g^1, ..., g^{N_v} \sim G(0, 1)$ are independent. We have that*

$$P(Y = k, V = k) = \frac{p^k q^k}{p^k q^k + \sum_{i \neq k} \max\{p^i q^k, q^i p^k\}}.$$

*Proof.* If

$$k = \arg\max_i\{\log p^i + g^i\} = \arg\max_i\{\log q^i + g^i\},$$

then we must have that for all $i \neq k$,

$$g^i < \log p^k + g^k - \log p^i \text{ and } g^i < \log q^k + g^k - \log q^i,$$

i.e.,

$$g^i < g^k + \min\{\log(p^k/p^i), \log(q^k/q^i)\} \quad \forall i \neq k.$$

Denoting by $F$ the CDF of the Gumbel distribution, we can write

$$P(Y = k, V = k | g^k) = \prod_{i \neq k} F(g^k + \min\{\log p^k/p^i, \log q^k/q^i\}).$$

Denoting the PDF of the Gumbel distribution by $f$ and marginalising we deduce that

$$P(Y = k, V = k) = \int_{-\infty}^{\infty} \prod_{i \neq k} F(g + \min\{\log p^k/p^i, \log q^k/q^i\}) f(g) dg.$$

Expanding, we can write this as

$$P(Y = k, V = k) = \int \prod_{i \neq k} \exp\{-\exp\{-g - \min\{\log p^k/p^i, \log q^k/q^i\}\}\} \exp\{-g - \exp(-g)\} dg$$

$$= \int \exp\{-g - \exp(-g) - \sum_{i \neq k} \exp\{-g - \min\{\log p^k/p^i, \log q^k/q^i\}\}\} dg$$

$$= \int \exp\{-g - \exp(-g)(1 + \sum_{i \neq k} \exp\{-\min\{\log p^k/p^i, \log q^k/q^i\}\}) dg.$$

Recall that

$$\frac{d}{dx} e^{e^x} = e^x e^{e^x} = e^{x+e^x},$$

and so for any $A$ independent of $x$ we have that

$$\frac{d}{dx} e^{-Ae^{-x}} = Ae^{-x} e^{-Ae^{-x}} = Ae^{-x-Ae^{-x}}.$$

Therefore, we may deduce that

$$P(Y = k, V = k) = \left[ \frac{1}{1 + \sum_{i \neq k} \exp\{-\min\{\log p^k/p^i, \log q^k/q^i\}\}} e^{-Ae^{-g}} \right]_{-\infty}^{\infty}$$

$$= \frac{1}{1 + \sum_{i \neq k} \exp\{-\min\{\log p^k/p^i, \log q^k/q^i\}\}}.$$

Since

$$\exp\{-\min\{\log p^k/p^i, \log q^k/q^i\}\} = \max\{\exp\{-\log p^k/p^i\}, \exp\{-\log q^k/q^i\}\}$$

$$= \max\{p^i/p^k, q^i/q^k\},$$

we deduce

$$P(Y = k, V = k) = \frac{1}{1 + \sum_{i \neq k} \max\{p^i/p^k, q^i/q^k\}}.$$

as claimed. $\qquad\square$

## B   Proofs relating to Gumbel recycling procedure

Note that in the following proofs, we denote for notational simplicity that for a random vector $x$, where each element of $x$ is independently sampled according to a Gumbel distribution, $x^k \sim G(0, 1)$, $p(x) = \prod_k G(x^k; 0, 1) = G(x; 0, 1)$.

## B.1 Overall proof

**Theorem B.1.** *Consider a sequence of tokens $Y_{1:T}$ generated auto-regressively according to the following update rule, where $k := \arg\max_j (g_t^j + \log h_t^j)$, $Q(\cdot)$ is the quantile function for the $G(0,1)$ distribution and $\pi_\theta(\cdot)$ is a language model:*

$$g_1 \sim G(0,1)$$
$$h_{t+1} = \pi_\theta(X, Y_{1:t})$$
$$g_{t+1}^k \mid g_t, h_t \sim G(0,1)$$
$$g_{t+1}^i \mid g_t, h_t = Q\left(\frac{Q^{-1}(g_t^i)}{Q^{-1}(g_t^k + \log h_t^k - \log h_t^i)}\right), \quad \text{for } i \neq k$$
$$Y_{t+1} = \arg\max_j \left(\log h_{t+1}^j + g_{t+1}^j\right)$$

*With this update procedure, the probability distribution over a given token conditioned on preceding tokens is*

$$p(Y_{t+1} \mid X, Y_{1:t}) = Cat(Y_{t+1}; h_{t+1})$$

*Proof.* We proceed through proof by induction. We make two assumptions that following expressions hold for $t$, then prove that the expressions hold for $t+1$ under those assumptions (and that they hold for the base case). The assumptions are that:

**Assumption 1.**
$$p(Y_t \mid X, Y_{1:t-1}) = Cat(Y_t; h_{t+1})$$

**Assumption 2.**
$$p(g_t \mid X, Y_{1:t-1}) = G(g_t; 0, 1)$$

N.B that in the base cases ($p(Y_1 \mid X)$ and $p(g_1 \mid X)$), the expressions are trivially valid by the Gumbel reparameterization trick and by construction of $g_1$ respectively.

Now, let's prove that the expressions hold for $t+1$. We will first prove the following:

$$p(g_{t+1} \mid X, Y_{1:t}) = G(g_{t+1}; 0, 1)$$

by first considering the probability $p(g_{t+1}^i < a \mid X, Y_{1:t})$:

$$p(g_{t+1}^i < a \mid X, Y_{1:t}) = \int p(g_{t+1}^i < a, g_t \mid X, Y_{1:t}) \, dg_t$$

$$= \int p(g_{t+1}^i < a \mid X, Y_{1:t}, g_t) p(g_t \mid X, Y_{1:t-1}, Y_t) \, dg_t$$

$$= \int p(g_{t+1}^i < a \mid X, Y_{1:t}, g_t) \frac{p(Y_t \mid X, Y_{1:t-1}, g_t) p(g_t \mid X, Y_{1:t-1})}{p(Y_t \mid X, Y_{1:t-1})} \, dg_t$$

$$= \frac{1}{p(Y_t \mid X, Y_{1:t-1})} \int p(g_{t+1}^i < a \mid h_t, Y_t, g_t) p(Y_t \mid h_t, g_t) G(g_t; 0, 1) \, dg_t$$

Firstly, consider the case where $Y_t = i$. In this case, we know that $g_{t+1}^i$ is newly sampled from $G(0,1)$. Therefore, using the Gumbel reparameterization trick for the last step, we have that:

$$p(g_{t+1}^i < a \mid X, Y_{1:t}) = \frac{1}{h_t^i} \int p(g_{t+1}^i < a \mid h_t, Y_t, g_t) p(Y_t \mid h_t, g_t) G(g_t; 0, 1) \, dg_t$$

$$= \frac{1}{h_t^i} \int Q^{-1}(a) p(Y_t \mid h_t, g_t) G(g_t; 0, 1) \, dg_t$$

$$= \frac{h_t^i}{h_t^i} Q^{-1}(a) = Q^{-1}(a)$$

Turning our attention to the case where $Y_t = j \neq i$

$$p(g_{t+1}^i < a \mid X, Y_{1:t}) = \frac{1}{p(Y_t = j \mid X, Y_{1:t-1})} \int p(g_{t+1}^i < a \mid h_t, Y_t = j, g_t) p(Y_t = j \mid h_t, g_t) G(g_t; 0, 1) \, dg_t$$

$$= \frac{1}{h_t^j} \int p(g_{t+1}^i < a \mid h_t, Y_t = j, g_t) p(Y_t = j \mid h_t, g_t) G(g_t; 0, 1) \, dg_t$$

We simplify notation by denoting the following events:

$$E' = \left\{ Q \left( \frac{Q^{-1}(g_t^i)}{Q^{-1}(g_t^j + \log h_t^j - \log h_t^i)} \right) < a \right\}$$

$$E_p = \left\{ g_t^p + \log h_t^p < g_t^j + \log h_t^j \right\}$$

Now, we can rewrite the following probabilities using these definitions:

$$p(g_{t+1}^i < a \mid h_t, Y_t = j, g_t) = \mathbf{1}_{E'(g_t)}$$

$$p(Y_t = j \mid h_t, g_t) = \left( \prod_{p \neq j} \mathbf{1}_{E_p(g_t)} \right)$$

$$p(g_{t+1}^i < a \mid X, Y_{1:t}) = \frac{1}{h_t^j} \int \mathbf{1}_{E'(g_t)} \left( \prod_{p \neq j} \mathbf{1}_{E_p(g_t)} \right) G(g_t; 0, 1) \, dg_t$$

Since $Q^{-1}(x)$ is a monotonic function, $E_i$ is equivalently defined as:

$$E_i = \left\{ Q^{-1} \left( g_t^i \right) < Q^{-1} \left( g_t^j + \log h_t^j - \log h_t^i \right) \right\}$$

Additionally, $E'$ can be rewritten as

$$E' = \left\{ Q^{-1} \left( g_t^i \right) < Q^{-1}(a) Q^{-1} \left( g_t^j + \log h_t^j - \log h_t^i \right) \right\}$$

Since $Q^{-1}(a) \in [0, 1]$, the occurrence of $E'$ is a sufficient condition for the occurrence of $E_i$. Therefore, we can simplify the integral to:

$$p(g_{t+1}^i < a \mid X, Y_{1:t}) = \frac{1}{h_t^j} \int \mathbf{1}_{E'(g_t)} \left( \prod_{p \neq i, j} \mathbf{1}_{E_p(g_t)} \right) G(g_t; 0, 1) \, dg_t$$

The CDF of the Gumbel distribution can be written $Q^{-1}(x) = e^{-e^{-x}}$, so $Q^{-1}(x+c) = \left( Q^{-1}(x) \right)^{e^{-c}}$. With this fact and application of the monotonic transformation $Q(\cdot)$, we can rewrite the events :

$$E' = \left\{ Q^{-1} \left( g_t^i \right) Q^{-1} \left( g_t^j \right)^{-\frac{h_t^i}{h_t^j}} < Q^{-1}(a) \right\}$$

$$E_p = \left\{ Q^{-1} \left( g_t^p \right) < Q^{-1} \left( g_t^j \right)^{\frac{h_t^p}{h_t^j}} \right\}$$

We now use the fact that $Q^{-1}(g_t^i) := U_t^i \sim \mathcal{U}[0, 1] \; \forall i$ to rewrite the events like so:

$$E' = \left\{ U_t^i \left( U_t^j \right)^{-\frac{h_t^i}{h_t^j}} < Q^{-1}(a) \right\}$$

$$E_p = \left\{ U_t^p < \left( U_t^j \right)^{\frac{h_t^p}{h_t^j}} \right\}$$

In conjunction with lemma B.1, this gives us the desired cumulative density function:

$$p(g_{t+1}^i < a \mid X, Y_{1:t}) = \frac{1}{h_t^j} (h_t^j) Q^{-1}(a) = Q^{-1}(a)$$

Since the cumulative density function in both cases ($Y_t = i$ and $Y_t \neq i$) is $Q^{-1}(a)$, we have that, under our initial assumptions, $p(g_{t+1} \mid X, Y_{1:t}) = G(g_{t+1}; 0, 1)$.

Finally, we then introduce and marginalise over the Gumbel noise vector at the previous timestep for the distribution over $Y_{t+1}$, where the final step follows from the Gumbel reparameterization trick:

$$
\begin{aligned}
p(Y_{t+1} \mid X, Y_{1:t}) &= \int p(Y_{t+1}, g_{t+1} \mid X, Y_{1:t}) \, dg_{t+1} \\
&= \int p(Y_{t+1} \mid X, Y_{1:t}, g_{t+1}) p(g_{t+1} \mid X, Y_{1:t}) \, dg_{t+1} \\
&= \int p(Y_{t+1} \mid h_{t+1}, g_{t+1}) G(g_{t+1}; 0, 1) \, dg_{t+1} \\
&= \mathrm{Cat}(Y_{t+1}; h_{t+1})
\end{aligned}
$$

Therefore, since the expressions are valid for the base case of $t = 1$, and we have shown them to be valid for $t + 1$ if assumptions 1 and 2 hold, they must be true for all $t$, by induction. $\qquad\square$

## B.2  Statement and Proof of lemma B.1

**Lemma B.1.** *$X$, $Y$ and $Z_{1:N}$ are random variables each independently drawn from $\mathcal{U}[0, 1]$. $A$, $B$, $C_{1:N}$ and $D$ are positive constants between 0 and 1, and $A + B + \sum_n C_n = 1$. Defining the events $E^* = \left\{ XY^{-\frac{A}{B}} < D \right\}$ and $E_n = \left\{ Z_n < Y^{\frac{C_n}{B}} \right\}$, the probability of the intersection of events is given by:*

$$
P\left( E^* \cap \bigcap_{n=1}^{N} E_n \right) = BD
$$

*Proof.* We can write down the following probabilities that are conditional on $Y$:

$$
P(E^* | Y) = P\left( X \leq DY^{\frac{A}{B}} \right) = DY^{\frac{A}{B}}
$$

$$
P(E_n | Y) = P\left( Z_n \leq Y^{\frac{C_n}{B}} \right) = Y^{\frac{C_n}{B}}
$$

Therefore, the probability of the complement is given by integrating the product of these quantities over $p(y)$:

$$
\begin{aligned}
P\left( E^* \cap \bigcap_{n=1}^{N} E_n \right) &= \int_0^1 P(E^* | Y) \prod_{n=1}^{N} P(E_n | Y) dY \\
&= \int_0^1 \left( DY^{\frac{A}{B}} \right) \prod_{n=1}^{N} \left( Y^{\frac{C_n}{B}} \right) dY \\
&= \int_0^1 \left( DY^{\frac{A + \sum_n C_n}{B}} \right) dY \\
&= D \frac{1}{\frac{A + \sum_n C_n}{B} + 1} \\
&= D \frac{1}{\left( \frac{A + \sum_n C_n + B}{B} \right)} = BD
\end{aligned}
$$

$\qquad\square$

## C  Proof of unbiased estimator for $p_{repeat}$

**Claim.** *Let $p$ denote the probability of some Bernoulli event. an unbiased estimator of $p$ given by a finite set $N$ of samples $Z_{1:N}$ from the distribution is given by:*

$$
\hat{p} = \frac{1}{N} \sum_{i=1}^{N} Z_i
$$

*An unbiased estimator of $p_{repeat} = p^2 + (1-p)^2$ is:*

$$\frac{N}{N-1}(\hat{p}^2 + (1-\hat{p})^2) - \frac{1}{N-1}$$

*Proof.* Calculate the expectation of $\hat{p}^2$:

$$\mathbb{E}(\hat{p}^2) = E\left(\left(\frac{1}{N}\sum_{i=1}^{N} Z_i\right)^2\right)$$

Expand the square inside the expectation:

$$\mathbb{E}(\hat{p}^2) = \frac{1}{N^2}\mathbb{E}\left(\sum_{i=1}^{N} Z_i^2 + \sum_{i\neq j} Z_i Z_j\right)$$

Since $Z_i^2 = Z_i$, and by linearity of expectation:

$$\mathbb{E}(\hat{p}^2) = \frac{1}{N^2}\left(Np + N(N-1)p^2\right)$$

Simplify the expression:

$$\mathbb{E}(\hat{p}^2) = \frac{Np + N^2 p^2 - Np^2}{N^2} = \frac{p + (N-1)p^2}{N}$$

Using this result, we have the following:

$$\begin{aligned}
\mathbb{E}(\hat{p}^2 + (1-\hat{p})^2) &= E\left(2\hat{p}^2 - 2\hat{p} + 1\right) \\
&= 2E(\hat{p}^2) - 2E(\hat{p}) + 1 \\
&= 2\left(\frac{1}{N}p + \frac{N-1}{N}p^2\right) - 2p + 1 \\
&= \frac{1}{N}\left((N-1)(2p^2 - 2p + 1) + 1\right) \\
&= \frac{N-1}{N}\left((2p^2 - 2p + 1) + \frac{1}{N-1}\right) \\
&= \frac{N-1}{N}p_{repeat} + \frac{1}{N}
\end{aligned}$$

Therefore, we can debias the naive estimator using the following expression:

$$\frac{N}{N-1}(\hat{p}^2 + (1-\hat{p})^2) - \frac{1}{N-1}$$

$\square$

# D   Justification for ensembling procedure

**Theorem.** *Suppose we have a set of categorical distributions $\{P_i\}_{i=1}^{n}$, define $Q^*$ as the distribution minimizing the average forward Kullback-Leibler divergence to each $\{P_i\}_{i=1}^{n}$:*

$$Q^* = \arg\min_{Q} \frac{1}{n}\sum_{i=1}^{n} D_{KL}(Q\|P_i) \tag{4}$$

*then $Q^*(x)$ can be expressed as*

$$Q^*(x) = \frac{1}{Z}\prod_{i=1}^{n} P_i(x)^{\frac{1}{n}} \tag{5}$$

*where Z is the normalisation constant to ensure $Q^*$ defines a valid probability distribution function*

$$Z = \sum_x \prod_{i=1}^n P_i(x)^{\frac{1}{n}}$$

*Proof.* Expanding the KL divergence

$$\frac{1}{n}\sum_{i=1}^n D_{\mathrm{KL}}(Q\|P_i) = \frac{1}{n}\sum_{i=1}^n \sum_x Q(x)\log\frac{Q(x)}{P_i(x)}$$

Changing the order of sums, this can be re-expressed as

$$\frac{1}{n}\sum_{i=1}^n D_{\mathrm{KL}}(Q\|P_i) = \frac{1}{n}\sum_x Q(x)\log\frac{Q(x)^n}{\prod_{i=1}^n P_i(x)} = \sum_x Q(x)\log\frac{Q(x)}{\prod_{i=1}^n P_i(x)^{\frac{1}{n}}}$$

Introducing the normalisation constant Z

$$\frac{1}{n}\sum_{i=1}^n D_{\mathrm{KL}}(Q\|P_i) = \sum_x Q(x)\log\frac{\frac{1}{Z}Q(x)}{\frac{1}{Z}\prod_{i=1}^n P_i(x)^{\frac{1}{n}}} = \frac{1}{n}\sum_{i=1}^n D_{\mathrm{KL}}(Q\|P_i)$$

separating the Z in the numerator

$$\frac{1}{n}\sum_{i=1}^n D_{\mathrm{KL}}(Q\|P_i) = \sum_x Q(x)\log\frac{Q(x)}{\frac{1}{Z}\prod_{i=1}^n P_i(x)^{\frac{1}{n}}} + \log\frac{1}{Z}$$

thus

$$\frac{1}{n}\sum_{i=1}^n D_{\mathrm{KL}}(Q\|P_i) = D_{\mathrm{KL}}(Q\|\frac{1}{Z}\prod_{i=1}^n P_i^{\frac{1}{n}}) + \log\frac{1}{Z}$$

this will be minimised when the right-hand side KL is equal to zero which occurs at $Q^*(x) = \frac{1}{Z}\prod_{i=1}^n P_i(x)^{\frac{1}{n}}$ $\qquad\square$

## E  Experimental details

For all experiments, answer generation is done using language models quantised to bfloat16 [32]. Chat and instruction-tuned models are prompted using default templates whereas base models (mistral, llama3-base) are prompted with a template consisting of a single in-context example to help steer away from off-topic answers. The addition of this in-context example was found to not materially impact the efficacity of Gumbel sampling but impact the quality of responses.

To avoid excessive experiment run-times, we restrict generated responses to a maximum length after which we interrupt text generation. This limit was set to 50 new tokens for semantic similarity experiments and to 200 new tokens for stylistic similarity experiments. We ran small-scale experiments with larger maximum response length and did not find material evidence of experimental findings being impacted by this response truncation.

When measuring semantic similarity between responses we measure consistency between responses rather than response-question pairs with any follow-up questions or answers hallucinated by the language model removed programmatically.

## F  Stylistic consistency prompts

### F.1  Aleatoric-list

Listing 1: Zero-shot classification prompt for whether model-generated response contains bulletpoints. Placeholders for question-specific content are shown in red.

```
1  Does the following response separate items in the answer using bullet
     points (*/-), letters (a/b/c...), numerics (1,2,3) or if items are not
     separated respond with 'doesn't separate '?
```

```
2
3  Response : ``` {response} ```.
4  To make your answer easy to extract respond with only one of the
        following options 'uses bullets '/'uses letters '/'uses numerics '/'
        doesn't separate '
```

Listing 2: Zero-shot classification prompt for whether model-generated response is terse. Placeholders for question-specific content are shown in red.

```
1  Is the response terse or not?
2  Response : ``` {response} ```.
3  To make your answer easy to extract respond with only one of the
        following options 'terse '/'not terse '.
```

## F.2    Code-Alpaca

Listing 3: Zero-shot classification prompt for determining programming language of model-generated response. Placeholders for question-specific content are shown in red.

```
1  What is the programming language used in the provided response. If no
        programming language is used return None
2  Response : ``` {response} ```.
3  Your response should only contain the answer and nothing else.
```

Listing 4: Zero-shot classification prompt for determining if model-generated response contains comments. Placeholders for question-specific content are shown in red.

```
1  Does the code snippet in the following response contain comments.
2  Response : ``` {response} ```.
3  Answer either with yes/no. Your response should only contain the answer
        and nothing else.
```

Listing 5: Zero-shot classification prompt for determining if model-generated response contains code snippet. Placeholders for question-specific content are shown in red.

```
1  Does the following response contain a code snippet.
2  Response : ``` {response} ```.
3  Answer either with yes/no. Your response should only contain the answer
        and nothing else.
```

# G  Example responses and semantic similarity

## G.1  Llama3-base-7B without gumbel consistency sampling at T=0.8

Table 3: Examples of semantic similarity between responses for various questions

| Question | Response 1 | Response 2 | Similarity (%) |
|---|---|---|---|
| Break down the application of word embeddings in Natural Language Processing / Illuminate the use of word embeddings in Natural Language Processing | Natural Language Processing(NLP) is a branch of Artificial Intelligence (AI) which deals with the interaction between computers and human languages. Word2vec is the most popular class of word embeddin... | It is used to represent words and their context with numerical values. | 68.0 |
| Expound on the function of a computer motherboard. / Illustrate the function of a computer motherboard. | The motherboard is the framework of the computer. It houses the CPU, RAM, and other components. | The computer's motherboard serves a number of functions. It directs information from the CPU to the memory, makes sure the proper voltage levels are reached, and directs the data flow in and out of th... | 80.5 |
| Decode the abbreviation DNA. / Share the meaning behind DNA. | DNA is an abbreviation for deoxyribonucleic acid. | DNA is also known as Deoxyribonucleic Acid. It is found in every single cell in the body and each cell contains the complete set of genetic information. It is the genetic blueprint that determines the... | 78.0 |

## G.2  Llama3-base-7B with gumbel consistency sampling (recycling) at T=0.8

Table 4: Examples of semantic similarity between responses for various questions

| Question | Response 1 | Response 2 | Similarity (%) |
|---|---|---|---|
| Break down the application of word embeddings in Natural Language Processing / Illuminate the use of word embeddings in Natural Language Processing | Embedding is the encoding of words in the context with a real-valued vector of a certain dimension. The low dimension helps to reduce the computational complexity of the models. Embedding can be learn... | Embedding is the encoding of words in the context of a sentence with a real-valued vector. The use of word embeddings helps to solve the problem of 'Word sense disambiguation'. | 85.0 |
| Expound on the function of a computer motherboard. / Illustrate the function of a computer motherboard. | The motherboard of a computer is responsible for providing a workspace where all the hardware components can be connected. The motherboard also controls the communication between the various component... | The motherboard of a computer is responsible for providing communication between all the various components. The motherboard also controls basic functions such as the timing of the hardware, start-up ... | 90.0 |
| Decode the abbreviation DNA. / Share the meaning behind DNA. | DNA stands for deoxyribonucleic acid | DNA means Deoxyribonucleic acid | 98.0 |

### G.3 Llama3-instruct-7B without gumbel consistency sampling at T=0.8

Table 5: Examples of semantic similarity between responses for various questions

| Question | Response 1 | Response 2 | Similarity (%) |
|---|---|---|---|
| Break down the application of word embeddings in Natural Language Processing / Illuminate the use of word embeddings in Natural Language Processing | Word embeddings are a fundamental component in Natural Language Processing (NLP) that allow machines to understand the nuances of language by mapping words to vectors in a high-dimen... | The wonders of word embeddings!

Word embeddings are a fundamental concept in Natural Language Processing (NLP) that revolutionized the way we represent words as vect... | 91.0 |
| Expound on the function of a computer motherboard. / Illustrate the function of a computer motherboard. | The motherboard is the main circuit board of a computer, connecting and supporting all the hardware components of the system. It serves as a communication highway, allowing different... | The computer motherboard is the main circuit board of a computer that connects all the hardware components together. It is responsible for facilitating communication and data transfe... | 91.0 |
| Decode the abbreviation DNA. / Share the meaning behind DNA. | A classic one!

The abbreviation DNA stands for:

D - Deoxy
N - Nucleic
A - Acid

In other words, DNA is Deoxyribose N... | What an fascinating topic! DNA (Deoxyribonucleic acid) is a molecule that contains the genetic instructions used in the development and function of all living organisms. It's often r... | 78.0 |

### G.4 Llama3-instruct-7B with gumbel consistency sampling (recycling) at T=0.8

Table 6: Examples of semantic similarity between responses for various questions

| Question | Response 1 | Response 2 | Similarity (%) |
|---|---|---|---|
| Break down the application of word embeddings in Natural Language Processing / Illuminate the use of word embeddings in Natural Language Processing | Embedding is the encoding of words in the context with a real-valued vector of a certain dimension. The low dimension helps to reduce the computational complexity of the models. Embedding can be learn... | Embedding is the encoding of words in the context of a sentence with a real-valued vector. The use of word embeddings helps to solve the problem of 'Word sense disambiguation'. | 85.0 |
| Expound on the function of a computer motherboard. / Illustrate the function of a computer motherboard. | The motherboard of a computer is responsible for providing a workspace where all the hardware components can be connected. The motherboard also controls the communication between the various component... | The motherboard of a computer is responsible for providing communication between all the various components. The motherboard also controls basic functions such as the timing of the hardware, start-up ... | 90.0 |
| Decode the abbreviation DNA. / Share the meaning behind DNA. | DNA stands for deoxyribonucleic acid | DNA means Deoxyribonucleic acid | 98.0 |

