# OpenReview forum: "Waste Not, Want Not; Recycled Gumbel Noise Improves Consistency in Natural Language Generation"
_NeurIPS.cc/2024/Workshop/SafeGenAi — SafeGenAi Poster_

### Official Review · Reviewer_vTXg · 2024-10-09
**This paper introduces Recycled Gumbel Consistency Sampling, a novel, efficient method that enhances the consistency of language model outputs across semantically similar prompts by introducing correlated noise, showing improvements in both semantic and stylistic consistency.**

**Rating:** 7
**Confidence:** 3

**Review:**

This paper proposes a novel sampling method, *Gumbel Consistency Sampling*, to improve the consistency of language model outputs across semantically similar prompts. The method introduces correlated noise during sampling to increase token overlap between responses. The authors demonstrate improvements in both semantic and stylistic consistency on several benchmarks and models.

**Pros**:

1. Proposes a novel, computationally efficient, and model-agnostic sampling method that improves language model consistency.
2. Provides a thorough theoretical analysis.
3. Demonstrates clear empirical improvements in both semantic and stylistic consistency across multiple models and benchmarks.
4. The paper is well-organized and clearly written.

**Cons**:

1. The evaluation is somewhat limited in scope, focusing primarily on short-form question answering. It would be better to see results on longer-form questions.
2. The paper does not include cost experiments for GCS and GCSwR to demonstrate the computational efficiency of the method.

---

### Official Review · Reviewer_QVAw · 2024-10-09
**Review for the Paper: "Waste not, want not; Recycled Gumbel noise improves consistency in natural language generation"**

**Rating:** 8
**Confidence:** 4

**Review:**

### **Summary**

This paper proposes a novel method for improving the consistency of language model outputs by leveraging the Gumbel reparameterization trick in the sampling process. The method, referred to as Gumbel Consistency Sampling (GCS) and Gumbel Consistency Sampling with Recycling (GCSwR), introduces a shared latent variable to generate correlated responses. The authors aim to reduce inconsistencies in the probabilistic sampling of language models, which often results in varied outputs for similar inputs. The approach improves semantic and stylistic consistency across different language models and tasks, demonstrating significant gains in benchmarks without additional training or performance degradation

### **Strengths**

1. **Strong Empirical Results**: The experiments demonstrate clear improvements in consistency metrics across several models and datasets, including semantic and stylistic dimensions. The gains in response similarity, particularly when using GCSwR, are substantial, reaching up to 10% in some cases
2. **Practical Application**: The method is computationally inexpensive and easy to implement, making it an appealing solution for practitioners seeking to improve the reliability of natural language generation (NLG) outputs without altering the underlying model architecture

### **Weaknesses**

1. **Stylistic Focus May Be Task-Dependent**: The paper emphasizes stylistic consistency, which may not be a critical requirement for all use cases. The importance of stylistic consistency varies depending on the task (e.g., coding-related tasks or factual Q&A may prioritize factual accuracy over style). The method’s utility for factual consistency in real-world applications remains unclear
2. **Lack of Long-Term Evaluation**: The paper primarily focuses on the short-term consistency of responses. Further exploration of how this method impacts the model's performance over extended conversations or complex multi-turn dialogues is needed to fully understand its applicability in more interactive scenarios

### **Detailed Comments**

- The paper offers a clear explanation of how the Gumbel reparameterization trick is used to improve consistency. By introducing a shared latent variable for correlated sampling, the method ensures that semantically similar prompts produce more consistent responses. This approach stands out for its simplicity and effectiveness, as it does not require changes to the model’s architecture or the underlying probabilities
- The choice to measure both semantic and stylistic consistency adds depth to the paper's evaluation. For instance, the improvements in response similarity on the Alpaca and Aleatoric-List datasets demonstrate the utility of the method in generating more coherent outputs across various language models. The use of stylistic dimensions such as whether code snippets contain comments or are formatted with bullet points further enhances the practical relevance of the method for tasks like code generation

---

### Official Review · Reviewer_gSUs · 2024-10-11
**Interesting approach to sampling**

**Rating:** 7
**Confidence:** 4

**Review:**

This paper proposes a new sampling approach for enhancing response consistency across different prompts. The key to their approach is conditioning on a latent variable at each step of the generation. They show that  interpreting Gumbel noise as a latent variable and conditioning sampling events on the same realisation of this latent variable increases the probability of selecting the same category with both distributions compared to sampling from each categorical distribution independently.

---

### Official Review · Reviewer_1k5m · 2024-10-12
**Theoretically grounded, need better experimentation**

**Rating:** 5
**Confidence:** 3

**Review:**

This paper offers an innovative and computationally efficient solution to the problem of improving consistency in LLMs by incorporating a latent variable into the next-token sampling process based on the Gumbel reparametrisation trick. The approach does not require additional training and is low on inference cost.


1. Quality: The authors provide formal proofs for their Gumbel reparameterization trick and back up their claims with quantitative results. The experimentation covers both semantic and stylistic consistency evaluation. the method is computationally efficient and can be applied without additional model retraining.
2. Clarity: Theory well explained. Paper is well-written, with clear explanations of the reparametrization trick the Gumbel Consistency Sampling
3. Originality: The use of the Gumbel reparametrization trick for improving consistency is a novel contribution. This method that doesn’t alter the model architecture or require retraining, fine-tuning or RLHF.
4. Significance: While there is no doubt that LLMs need to generate consistent response to maintain user trust and increase reliability, the paper could have motivated this better by calling out explicit use cases where consistency is responses is crucial.

Pros: As covered above - novelty of contribution,  experiments on multiple LLMs which shows performance improvements, computational efficiency and very clear theoretical proof.

Cons:
1. The paper's explanation of semantic similarity could be clearer. The reliance on the E5mistral-7b model for measuring semantic similarity is mentioned, but the underlying mechanics of how this model functions and why it was chosen over other methods (like sentence-bert for example) are not fully explored.  Just using one evaluation method raises the question of whether the results generalize to other similarity evaluation methods.
2. One of the primary motivations foe paper is significant variance of LLM responses in style, factual accuracy, and tone. However, more attention is given to stylistic eval and less to factual accuracy and tone consistency. Infact the semantic evaluation does specifically talk about factual accuracy or tone consistency.
3. Because the method is measuring token similarity, and looking at responses in appendix, it does seem like the there is potential over-smoothening. Authors should sufficiently explore the trade-off between consistency and diversity. Maybe also extend the evaluation to cover how the method performs on diversity specific benchmarks. Also curious to know how it performs over long formats of text (rephrasing, stories/essay writing since it's harder to maintain consistency over long responses)

nit: There is no explicit mention of if the rephrased questions using GPt-4o mini were also measured for semantic similarity before using them in experiments. Authors should clarify this.